# [Re] Explaining Groups of Points in Low-Dimensional Representations

**Damiaan J.W. Reijnaers**
info@damiaanreijnaers.nl

**Daniël B. van de Pavert**
daniel.vandepavert@student.uva.nl

**Giguru Scheuer**
giguru.scheuer@student.uva.nl

**Liang Huang**
liang.huang@student.uva.nl

## Reproducibility Summary

**Scope of Reproducibility**

In this paper we present an analysis and elaboration of (6), in which an algorithm is posed by Plumb *et al.* for the purpose of finding human-understandable explanations in terms of given explainable features of input data for differences between groups of points occurring in a lower-dimensional representation of that input data.

**Methodology**

We have upgraded the original code provided by the author such that it is compatible with recent versions of popular deep learning frameworks, namely the `TensorFlow 2.x`- and `PyTorch 1.7.x`-libraries. Furthermore, we have created our own implementation of the algorithm in which we have incorporated additional experiments in order to evaluate the algorithm's relevance in the scope of different dimensionality reduction techniques and differently structured data. We have performed the same experiments as described in the original paper using both the upgraded version of the code provided by the author and our own implementation taking the authors' code and paper as references.

**Results**

The results presented in (6) were reproducible, both by using the provided code and our own implementation. Our additional experiments have highlighted several limitations of the explanatory algorithm in question: the algorithm severely relies on the shape and variance of the clusters present in the data (and, if applicable, the method used to label these clusters), and highly non-linear dimensionality reduction algorithms perform worse in terms of explainability.

**What was easy**

The authors have provided an implementation[1] that cleanly separates different experiments on different datasets and the core functional methodology. As a result of this separation, given a working environment, one could easily reproduce the experiments performed in (6).

**What was difficult**

Minor difficulties were experienced in setting up the required environment for running the code provided by Plumb *et al.* locally (i.e. trivial changes in the code such as the usage of absolute paths and obtaining external dependencies). Evidently, it was time-consuming to rewrite all corresponding code, including the architecture for the variational auto-encoder provided by an external package, `scvis 0.1.0`[2].

**Communication with original authors**

No communication with the original authors was required to reproduce their work.

---

[1]GitHub, *Explaining Low Dim. Representations*, `https://github.com/GDPlumb/ELDR`, accessed on January 29th, 2021

[2]GitHub, *shahcompbio/scvis: Python package for dimension reduction of high-dimensional biological data*, `https://github.com/shahcompbio/scvis`, accessed on January 22nd, 2021

# 1 Introduction

As AI models are getting more integrated into applications with economic or social implications, the need for *explaining* decisions made by (potentially complex) models is increasing. In many AI applications, big datasets form the basis of a decision-making algorithm (5). As these datasets often involve data of high dimensionality, the dimensionality of data is, in many different applications, often reduced using *dimensionality reduction* (DR) techniques (10).

Data, and consequently the decisions made by an algorithm that are (in)directly based on that data, often involve some sort of 'grouping,' either by utilizing a *clustering method* or by (manually) pre-defined 'clusters of data.' The 'grouping process' may happen before the dimensionality reduction, for which, assuming well-organized data in which the dimensions correspond to explainable 'real-life' *features* of the phenomena measured in the data, the distinguishing characteristics of a certain group becomes relatively explainable as these groups are marked by boundaries in terms of explainable dimensions. However, this process can also happen after the dimensionality of the data has been reduced, which results in the groups being defined in terms of dimensions in a *latent space*. Especially when grouping occurs after dimensionality reduction, different clusters in data often play a key role in the decision-making process of an algorithm – one could, for example, when regarding *credit risk modelling*, point out a group in latent space which 'poses a high risk' when provided a loan (4). Moreover, such groupings arise naturally under the context of (variational) auto-encoders, which are the architecture of preference for many deep learning applications (6, p. 8).

Thus, considering the above-mentioned need for explainable machine decisions, in combination with the increasing use of DR algorithms and the reliance on observed data clusters in abstract latent spaces which are not understandable to the human mind, it is of great relevance and importance to develop a method to explain differences between different groups in the light of the application of a certain dimensionality reduction technique.

Elaborating upon the earlier introduced example of credit risk, one could argue that a reasonable explanation for a machine decision is of the kind: *"Your loan was rejected, but if you would have earned €422,86 more per month, your loan would have been accepted."*. This type of explanation is a *counterfactual explanation* – a decision on the basis of a 'fake,' counterfactual, 'world,' in which features *would have* been shaped differently. The proposed explanatory technique in (6) does exactly this: it reverse-engineers the obtained cluster labels in latent space to label the corresponding data in original space; then 'tweaks' an initial cluster by translating the initial cluster in original space, so that it is mapped to (approximately) the same point in latent space. Since the dimensions in original space correspond to explainable features—the characteristics of a certain group—which are comprehensible for a human mind; a translation that corresponds to merely adding or subtracting values to each of these features, can be perfectly explained in 'human language.' Since it is desirable for the explanations to be concise, the translation should be as *sparse* as possible: the translation should 'tweak' as few dimensions as possible. The intuition behind this idea is visualized in figure 1.

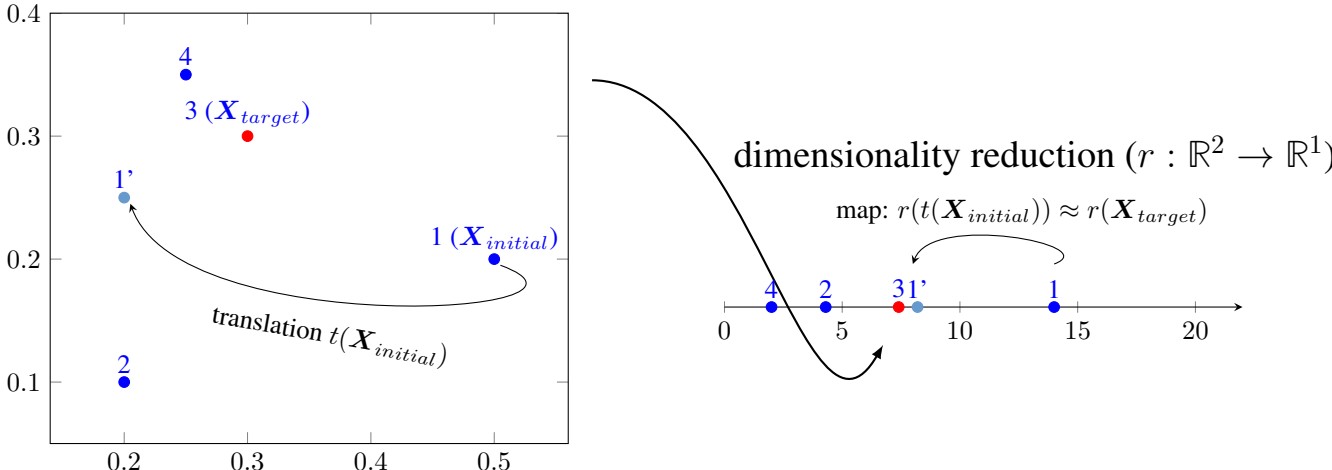

Figure 1: Visualisation of the method described by Plumb *et al*. Each point represents an individual cluster. The difference between clusters 1 and 3 are explained. Note that although point 1 maps to $1'$, which is far from point 3 in original space, point $1'$ is almost identical to 3 in latent space – this is the goal, as we want to find a point $1'$ which maps 1 as close to 3 in latent space while keeping the translation from 1 to $1'$ as sparse as possible. Note that the shown clusters (1-4) are determined in latent space and back-engineered to original space. $r(\boldsymbol{X}_i)$ is also referred to in (6) as $\boldsymbol{R}_i$.

## 2 Scope of reproducibility

As follows from the introduction, the authors of (6) opted for a counterfactual, sparse explanation for key differences between (naturally arising) groups. Plumb *et al.* propose an algorithm that generalizes this idea and attempts to find *global* differences between all groups, constructed through a composition of simpler explanations and introduced as *Transitive Global Translations* (TGT). The main tested contributions are that "TGTs provide a *global* and *counterfactual* explanation between all groups which is mathematically consistent (i.e. symmetrical and transitive)." and that "TGTs overcome the shortcomings of statistical and manual interpretation which do not use the model that learned the low dimensional representation that was used to define the groups in the first place."

In addition to simply replicating these claims using the provided codebase, we further evaluated these statements by testing compatibility with other DR algorithms than the originally used variational auto-encoder and tested the applicability of the algorithm on different datasets with different internal structures. All of the latter mentioned experiments were performed by rewriting and implementing the algorithm in `PyTorch` (taking the authors' code and paper as reference), whereas the original code is written using the `TensorFlow`-framework, motivated by the development of `PyTorch` in becoming the preferred framework by researchers (3) and the fact that it offers clearer coding structure[3]. For the sake of ensuring future reproducibility of the experiments originally presented in (6), as an addition, we have also provided an upgraded version of the original code, without further modifications, which is now compatible with more recent versions of `TensorFlow`.

## 3 Methodology

In accordance with section 2, for the purpose of reproducing (6), several steps were taken in order to best cover the scope of the original paper with limited resources. Firstly, the provided artifacts were 'modernized.' Concretely – the code provided by Plumb *et al.* and the external library scvis were upgraded to be compatible with `TensorFlow 2.4.1`[4] and `matplotlib 3.3.3`[5]. Secondly, the provided code was rewritten from scratch – both the code complementing the original paper; and the scvis library, to make the computation of the explanations independent from the DR algorithms. Thirdly, we have run our implementation on different DR algorithms—both linear and non-linear (see section 3.1)—and on different datasets (see section 3.3). Both the reproduced version (using original code) and the rewritten version are further explained in section 3.2. All relevant code complementing (6)—the chunks which we have chosen to rewrite—are explicitly stated in the paper, making the method by Plumb *et al.* reproducible, even if the code would not have been provided by the authors.

### 3.1 Dimensionality reduction algorithms

Throughout (6), DR algorithms are only mentioned in the general sense. Only in section 4 the algorithm which Plumb *et al.* use for their experiments is introduced: a variational auto-encoder (VAE), which is a non-linear family of dimensionality reduction algorithms based on (deep) symmetrical neural networks with a *bottleneck* in the middle layers which form the latent representation of the input data. The implemented VAE is based on the architecture proposed by Ding *et al.* (2).

The explanatory method presented by Plumb *et al.* *should* work for any DR algorithm while effectively treating the algorithm as a 'black box.' Apart from testing this hypothesis, it is relevant to compare 'explanatory performance' on different DR algorithms as different algorithms suit different types of data. Furthermore, an explanation based solely on translations could possibly perform worse in situations wherein data is transformed in a non-linear manner. Especially when methods inherently non-linearly transform (and warp) the input space. Therefore, as an addition to simply reproducing the implementation of Plumb *et al.*, experiments were done with several commonly known DR algorithms listed below.

#### 3.1.1 Linear methods

We have opted for two different linear DR algorithms: truncated SVD (TSVD) and sparse PCA (SPCA) (11). Both TSVD and SPCA reduce the dimensionality in a linear fashion. However, a key difference between both algorithms is

---

[3]Medium, `https://towardsdatascience.com/pytorch-vs-tensorflow-spotting-the-difference-25c75777377b`, accessed on January 25th, 2021

[4]TensorFlow, *Automatically upgrade code to TensorFlow 2*, `https://www.tensorflow.org/guide/upgrade`, accessed on January 22nd, 2021

[5]Matplotlib, *API Changes*, `https://matplotlib.org/3.3.3/api/api_changes.html`, accessed on January 22nd, 2021

that SPCA centers the data before computing the decomposition, where TSVD does not. As a result, TSVD handles sparse data more efficiently.

The objective of sparse PCA is to find the sparse components that best reconstruct the data. While standard PCA, in most cases, extracts components using dense expressions, these are often hard to interpret. However, the sparse vectors extracted by sparse PCA naturally match the latent components, which increases explainability.

### 3.1.2 Non-linear methods

The already mentioned VAE implementation is based on a Gaussian distribution, which results in a probabilistic generative model that preserves both local and global neighbor structures in the data (2). As the VAE is solely used for encoding a constant latent space, the model is trained on the entire dataset. We further incorporate the use of three additional non-linear methods: kernel PCA (KPCA), locally linear embedding (LLE) (7) and isomaps. The latter two mentioned DR techniques are examples of *manifold learning* (1).

KPCA (8) is a non-linear variant of PCA, which achieves a non-linear transformation of the input space by extending standard PCA through the use of kernels. These kernels effectively mimic a complex function that projects the input data on a *higher* dimensional space in which, consequently, a lower-dimensional subspace is found in which the data is represented, resulting in a more efficient low-dimensional latent representation. The advantage is that KPCA is able to identify clusters which are not linear separable. For our experiments, we found that a sigmoid kernel provided the best performance (measured by the metrics proposed in section 3.2) when the latent space resulting from the KPCA was used to find TGTs.

Finally, manifold learning techniques are implemented to analyse whether TGTs can still perform on a higher-dimensional embedding of a low-dimensional manifold, especially for datasets of which its data might not lay on an underlying low-dimensional manifold. Isomaps can be seen as an extension of KPCA. Isomaps present a lower dimensional space while maintaining distances between all points, while LLE aims to map to a lower dimensional space while maintaining the distance in local neighborhoods. While Isomap could be seen as an extension of KPCA, LLE can be understood as a combination of PCAs run on local neighborhoods.

As shown in section 4.1, different latent spaces (resulting from applying different processes for the purpose of reducing dimensions) seem to perform better in terms of explainability, which is further discussed in section 5.

### 3.2 Model descriptions

The original paper aimed to find an explanation for the key differences between a pair of groups in latent space where the explanation is expressed in terms of the original dimensions. This explanation is represented by a counterfactual translation of one of the groups (in the pair) in original space: "*what if* all points in group A, thus $\forall x \in A, x \in \mathbb{R}^d$, would have been translated, so that $\forall x \in A, \delta \in \mathbb{R}^d, x' = x + \delta$? Would group A have been roughly the same as group B *after* the groups have been mapped to latent space, so that $\forall x \in A, \forall y \in B, x \in \mathbb{R}^d, y \in \mathbb{R}^m, r : \mathbb{R}^d \to \mathbb{R}^m, r(x + \delta) \approx r(y)$?".

As both the original space ($\mathbb{R}^d$) and latent space ($\mathbb{R}^m$) are Euclidean spaces, translations between any pair of groups within a larger number of groups ($> 2$) can be constructed by composing two translations (a vector addition in this context ) or negating a translation (in this context equivalent to vector-scalar multiplication with $\lambda = -1$). Using these operations and a *reference group*, explanations ($\delta$) can be obtained for every possible pair of groups. The intuition behind this idea is illustrated in figure 2. Points in figure 2 do not directly correspond to the groups for which we want to find differences, since these locations will be approximated using sparse translations (see figure 1). Moreover, this linearity enables the possibility to measure the difference between the points in the two groups in latent space using the $l_2$-norm of the squared differences. At the same time, it is also possible to measure the *sparsity* of $\delta$ using $l_1$-regularization, which directly relates to the comprehensibility of the explanation posed by $\delta$.

In order to obtain adequate components for the $\delta$-vector between a group and the reference group, all $\delta$-vectors can be initialized to zero and optimized by adjusting the components of $\delta$ corresponding to two randomly chosen groups in the negative direction of the gradient of a loss function which accounts for the above-mentioned constraints. This loss function is formulated in equation 9 of the original paper. While Plumb *et al.* decide to incorporate the calculation of the gradient as an 'extension' of the `scvis` package, which we have reproduced using the provided code, we have decided to analytically compute the gradient using `SciPy 1.6.0` (using the `optimize.approx_fprime`-method). This implementation choice is in line with our aim to rewrite all code from scratch to obtain a 'stand-alone' division between the explanatory algorithm (which is the main idea of the original paper) and various 'black-box' dimensionality reduction algorithms, including the variational auto-encoder (the only algorithm Plumb *et al.* experiment with), as introduced in the beginning of this section.

The model performance is measured by two separate, but related (see section 4.1), metrics defined by (6, p. 3): *correctness*, which measures the degree to which projected points are actually *in the vicinity* of points they should map to (which they, in a sense, should 'imitate') and *coverage*, which measures the degree to which projected points *cover* the target group. Both are defined in equation 3–4.

### 3.3 Datasets

In the upgraded-code model, we decided to reproduce the explanations on all provided datasets, including the synthetic and corrupted versions. The datasets used in the experiments in the original paper are the Heart Disease, Boston Housing, and Iris datasets; and the single-cell RNA dataset (9). Our from-scratch model has all provided datasets incorporated, except for the latter, due to lack of computation power. However, experiments were run on three additional UCI datasets: Seeds, Wine and Glass[6]. The additional sets allow us to further understand how varying underlying structures in data influence explainability using different DR algorithms. Not all datasets can be reduced by all techniques: all three added datasets do not yield eigencomponents for a sigmoidal Kernel-PCA procedure.

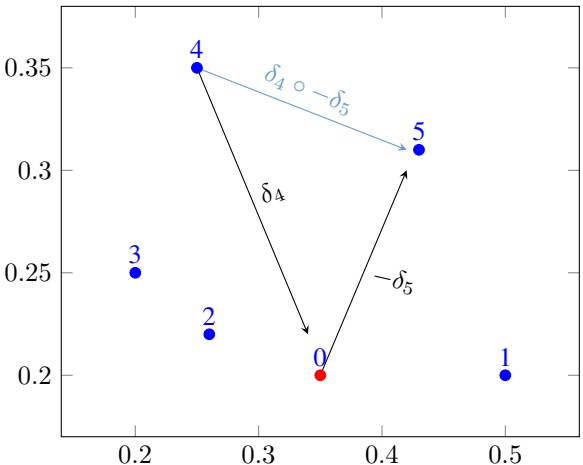

Figure 2: Visualisation of the intuition behind the composition of two translations with regard to a reference group 0, of which one is negated. Note that in this example every data point represents a whole group.

### 3.4 Hyperparameters

A constraint on the resulting explanations is that they must be sparse. The sparsity level is formally defined by $k$, representing the number of dimensions in the original space (with 'real-life' features) used as the main components of the translation that forms the explanation. The hyperparameter $k$ is enforced *after* the learning process by truncating the $\delta$-vector. To enforce sparsity *during* the learning process, the $\delta$-vector is being $l_1$-regularized by a term $\lambda$, as follows from (6, p. 5) and was inspired by the field of *compressed sensing*. The resulting equation, which is minimized, is stated in equation 1. As further explained in section 3.5, different values for $\lambda$ are tested for optimization.

$$loss(\delta) = ||r(\bar{x}_{initial} + \delta) - \bar{r}_{target}||_2^2 + \lambda||\delta||_1 \tag{1}$$

Following (6, p. 6), a *similarity measure* is used to capture the degree of which the features of a $k_1$-explanation also correspond to $k_1$ dimensions of a $k_2$-explanation where $k_2 > k_1$. This measure is defined in equation 2.

$$similarity(e_1, e_2) = \frac{\sum |e_1[i]|\mathbb{1}[e_2[i] \neq 0]}{||e_1||_1} \tag{2}$$

Lastly, hyperparameter $\epsilon$ defines a threshold for the *correctness* and *coverage* metrics, as defined in equation 3–4. Although Plumb *et al.* do hint on which type of search they have performed to find values for $\epsilon$, it is not clearly expanded upon (6, p. 4). For the purpose of analysing reproducibility, we have adopted the same values for $\epsilon$ (which were not present in the paper) for the same datasets. For all new datasets, we have introduced similar static values for $\epsilon$. Plumb *et al.* do, however, propose an evaluation method for $\epsilon$, although it is again not mentioned in the original paper. The authors take the minimum, maximum and mean value of the diagonal of the matrix of the correctness measures for all clusters (similar to the matrices shown in figure 4). These values might be misleading, as larger values for $\epsilon$ would inherently scores high on correctness. Therefore, we have constrained $\epsilon \in [0, 2]$ and dynamically scale the latent spaces to force the data to be spread out (see section 3.5). We have evaluated all values for $\epsilon$ and outline the means in table 1.

$$correctness(t) = \frac{1}{|\boldsymbol{X}_{initial}|} \sum_{x \in \boldsymbol{X}_{target}} \mathbb{1}[\exists x' \in \boldsymbol{X}_{target} |||r(t(x)) - r(x')||_2^2 \leq \epsilon] \tag{3}$$

$$coverage(t) = \frac{1}{|\boldsymbol{X}_{target}|} \sum_{x \in \boldsymbol{X}_{initial}} \mathbb{1}[\exists x' \in \boldsymbol{X}_{initial} |||r(x) - r(t(x'))||_2^2 \leq \epsilon] \tag{4}$$

---

[6]UCI Machine Learning Repository, `https://archive.ics.uci.edu/ml/datasets/{heart+disease,iris,seeds,Wine,glass+identification}` and `https://archive.ics.uci.edu/ml/machine-learning-databases/housing/`, accessed on January 29th, 2021; note the set notation in the latter section of the URL.

|          | VAE  | PCA  | TSVD | KPCA | SPCA | ISO  | LLE  | $|clusters|$ | $\epsilon$ |
|----------|------|------|------|------|------|------|------|------|------|
| Housing  | 0.99 | 1.00 | 1.00 | 1.00 | 1.00 | 1.00 | 0.83 | 6 | 1.50 |
| Iris     | 0.98 | 0.96 | 0.98 | 0.94 | 0.96 | 0.85 | 0.99 | 3 | 0.75 |
| Heart    | 0.96 | 0.99 | 1.00 | 0.99 | 0.99 | 0.98 | 1.00 | 8 | 1.00 |
| Seeds    | 1.00 | 0.97 | 1.00 | -    | 0.97 | 0.98 | 0.94 | 3 | 1.00 |
| Wine     | 1.00 | 0.99 | 1.00 | -    | 0.99 | 0.99 | 1.00 | 3 | 1.00 |
| Glass    | 0.99 | 0.94 | 0.94 | -    | 0.94 | 0.94 | 0.98 | 7 | 1.75 |

Table 1: Evaluation of fixed epsilon values present in equations 3–4: mean of diagonal of correctness matrix

### 3.5 Experimental setup and Computational requirements

We have reproduced the experiments presented in the paper by Plumb *et al.* by using the upgraded `TensorFlow`-code. Additionally, we have run our from-scratch code on three additional datasets and six other dimensionality reduction algorithms. All experiments were run for five trials for eleven different values of $\lambda$ (evenly spaced between 0 and 5), for which the best combination of $\delta$-vectors is chosen for every different value of $k$ ($k$ is evenly spaced between 1 and the number of dimensions $d$ in the original space, with a step size of 1 for $d \leq 5$, and 2 otherwise).

As pointed out in the previous section, an inadequately high value for $\epsilon$ poses a problem if the data in latent space is of low variance (especially if variance $< \epsilon$), as the performance measures, introduced in section 3.2 and shown in equations 3–4, would unjustly report very high scores. We have solved this problem by rescaling the data in latent space so that a variance of 10 is preserved (which amounts to a standard deviation of $\approx 3.16$). We have deliberately chosen not to utilize a method for removing *outliers* prior to defining the factor with which to rescale the data in latent space, as to preserve 'the spirit of the data,' especially since the internal structure of some datasets contain outliers which potentially correspond to groups with a significantly different character, as opposed to outliers resulting from noisy measurements. We further discuss this choice in section 5. However, as we failed to preserve an adequate amount of variance when performing *TSVD* (see section 3.1) in the *Glass*-dataset (see section 3.3) using this method, we manually scaled the data in this latent space, for this particular dataset, with an additional factor of 20.

In the from-scratch implementation, K-means is used for clustering, while for the replicated experiments in the original notebooks, following the code provided by the authors, clusters were manually selected.

Experimental meta-results are shown in table 2. The original code, the 'modernized' version of the original code and the code for the from-scratch implementation can be found on our GitHub repository[7].

|                          | VAE + model | PCA | TSVD | KPCA | SPCA | ISO | LLE | $|k|$ | $|clusters|$ |
|--------------------------|-------------|-----|------|------|------|-----|-----|-----|------|
| Housing $\in \mathbb{R}^{13}$ | 3.7 + 0.1   | 0.3 | 0.3  | 4.4  | 0.4  | 4.5 | 4.7 | 7 | 6 |
| Iris $\in \mathbb{R}^{4}$     | 0.8 + 0.1   | 0.1 | 0.1  | 0.7  | 0.1  | 1.0 | 1.1 | 4 | 3 |
| Heart $\in \mathbb{R}^{14}$   | 2.6 + 0.1   | 0.6 | 0.5  | 4.9  | 0.5  | 4.3 | 3.6 | 7 | 8 |
| Seeds $\in \mathbb{R}^{7}$    | 0.4 + 0.1   | 0.2 | 0.1  | -    | 0.2  | 3.0 | 2.7 | 4 | 3 |
| Wine $\in \mathbb{R}^{13}$    | 1.6 + 0.1   | 0.2 | 0.2  | -    | 0.2  | 4.2 | 4.8 | 7 | 3 |
| Glass $\in \mathbb{R}^{10}$   | 0.9 + 0.1   | 0.4 | 0.3  | -    | 0.3  | 2.8 | 2.8 | 5 | 7 |

Table 2: Rounded TGT training time in hours per dataset and DR algorithm for all values of $k$, measured on an i7-4720HQ CPU @$\approx$ 2.6GHz. Measured for 5 trials per $\lambda \in \{0, 0.5, \ldots, 5\}$. VAE models train on min. 3000 iters.

## 4  Results

Using our upgraded `TensorFlow`-code and the provided pre-trained VAE models, the obtained results are identical to those provided by the codebase of the paper for all datasets and methods[8]. After retraining all VAE models and re-learning all explanations, using the same upgraded `TensorFlow`-code, we obtain very similar results[9]. The results are not identical, as the models learn slightly different representations, in which the manually selected clusters also differ. This indicates that the used cluster generation techniques influence the model's ability to explain based on a translation. An example arises with the Iris dataset: Plumb *et al.* yield 0.833 coverage, while we get only 0.66. This

---

[7]`https://github.com/damiaanr/fact-ai`

[8]To compare, open the *.ipynb* in all folders (except 'code,' 'scvis,' 'MiscFigures,' and 'Integrated-Gradients-master') on both repositories, and look at the resulting plots: `https://github.com/GDPlumb/ELDR` and `https://github.com/damiaanr/fact-ai/tree/main/ELDR-TF2.x_(pre_trained_models)`

[9]Now, compare the plots with `https://github.com/damiaanr/fact-ai/tree/main/ELDR-TF2.x_(newly_trained_models)`

difference caused by a different organization of clusters propagates further into the results for the corrupted data. Except for minor differences arising from this same issue, we have successfully reproduced the results for all other datasets.

When running the Housing, Iris and Heart datasets on the from-scratch implementation of the explanation algorithm using our implementation of scvis, we obtain either similar (see figure 3) or even better results than Plumb *et al.* do. This again indicates that the method for dimensionality reduction does impact the results.

For different datasets and different dimensionality reduction algorithms that map the original datasets to different latent spaces, we have encountered the same problem related to the inability of explaining pairs of groups of which the clusters have a different standard deviation, as shown in the original paper in figures 4-7 (6, p. 4). This problem occurs, among others, in figure 5f in Appendix A (and the corresponding measures in figure 6f).

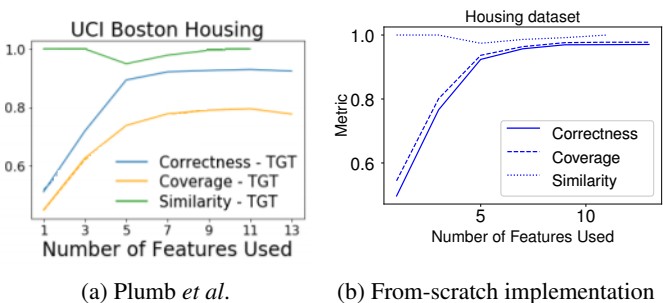

(a) Plumb *et al.*    (b) From-scratch implementation

Figure 3: Comparison of results for Housing data on VAE

### 4.1 Additional results not present in the original paper

As (6, p. 6) pointed out, the *correctness* or *coverage* metrics are closely related. If the translations between arbitrary groups are symmetrical, the former translation—which is the technical representation of the explanation—is the negative of the latter (and vice-versa). Thus, when applying different **linear** dimensionality reduction algorithms, both metrics will exactly equal each other, as follows from the results shown below in table 3, and in figure 6 (along all different values for $k$, the graphs conveying correctness and coverage are on top of each other for all linear algorithms) and figure 4 (note the color map plot for correctness is the 'transpose' of the plot for coverage, and vice-versa).

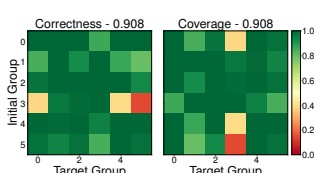

Figure 4: Metrics plot for Housing on PCA, $k = 5$

An additional observation is that topology-based dimensionality reduction techniques are less suited for data compression when the compressed data consequently needs to be explained using translation-based counterfactual explanations. Locally linear embedding and isomapping are reduction methods based on manifold mapping: a one- or two-manifold is embedded in two-dimensional space, and points of higher-dimensional space are mapped onto this manifold. This often yields problematic data structures. For instance, when data is represented using a line (a one-manifold) and embedded in two-dimensional space in which the one-dimensional line is twirled. Since we are only considering translations (and, *e.g.*, no rotations), the method is unable to adequately explain differences between clusters present on manifolds in latent space.

Furthermore, we have observed that certain datasets might yield constant results when varying the sparsity-constraint. In this case, the compression from high-dimensional to two-dimensional space was (nearly) *lossless* – a single component suffices to explain the difference between any group. This phenomenon is shown in figure 6e in Appendix A, indicating that the chemical composition of wine actually depends on only one latent variable: when using PCA, $99.98\%$ of the variance is explained by only two components (of which $99.81\%$ by the first component). Note that the input space spans 13 dimensions. It is also interesting to note that the VAE learns an approximation of LLE space (see figure 6e).

All results for all introduced performance measures, datasets and dimensionality reduction algorithms, are shown in table 3 below and illustrated in figure 5–6 in Appendix A. Sample explanations are shown in figure 7.

|  | VAE | PCA | TSVD | KPCA | SPCA | ISO | LLE |
|---|---|---|---|---|---|---|---|
| Housing | 0.97; 0.98; 0.99 | 0.99; 0.99; 0.99 | 1.00; 1.00; 0.99 | 0.99; 0.98; 0.99 | 0.99; 0.99; 0.99 | 0.22; 0.22; 0.99 | 0.20; 0.25; 1.00 |
| Iris | 0.96; 0.94; 1.00 | 0.89; 0.89; 1.00 | 0.95; 0.95; 1.00 | 0.82; 0.86; 1.00 | 0.88; 0.88; 1.00 | 0.67; 0.66; 1.00 | 0.54; 0.55; 1.00 |
| Heart | 0.83; 0.82; 0.98 | 1.00; 1.00; 0.98 | 1.00; 1.00; 0.98 | 0.99; 0.99; 0.98 | 1.00; 1.00; 0.98 | 0.16; 0.17; 0.98 | 0.40; 0.51; 1.00 |
| Seeds | 0.40; 0.40; 0.99 | 0.95; 0.95; 0.99 | 0.84; 0.84; 0.99 | - | 0.95; 0.95; 1.00 | 0.93; 0.94; 1.00 | 0.88; 0.87; 1.00 |
| Wine | 0.39; 0.39; 0.33 | 0.40; 0.40; 0.33 | 0.52; 0.52; 0.33 | - | 0.39; 0.39; 0.33 | 0.39; 0.39; 0.33 | 0.37; 0.37; 0.33 |
| Glass | 0.33; 0.33; 0.86 | 0.82; 0.82; 0.86 | 0.94; 0.94; 0.86 | - | 0.82; 0.82; 0.86 | 0.58; 0.61; 1.00 | 0.33; 0.45; 1.00 |

Table 3: Correctness, coverage and similarity scores respectively for each model and dataset. Mean value is shown for the similarity scores while the score for the largest $k$ is taken for the correctness and coverage measures.

# 5  Discussion

We have been able to successfully reproduce and upgrade the code provided by Plumb. *et al*, which is the implementation of the technique presented in (6). We were able to reproduce results by running the pre-loaded models; after re-training these models; and by rewriting the algorithm from scratch.

The performance depends on the mapping to latent space. The limitation of the algorithm is the lack of variable freedom: only translations can be used to explain differences between groups. By utilizing different DR methods, we have shown that not all algorithms produce latent spaces in which translations suffice for an explanation between clusters. Especially manifold-based algorithms require a more sophisticated type of explanation using, for instance, rotation and scaling.

A possible solution would be to generate an explanation in terms of a matrix (so that $x' = Mx + \delta$ instead of $x' = x + \delta$). However, explanations using translations are directly interpretable for humans as the dimensions of our datasets correspond to explainable features. Considering, for example, rotations, would come at the cost of explainability. However, forcing $M$ to be diagonal (which leads to multiplying all data features by corresponding factors) might yield proper explanations (of the type *"If your monthly income would have been twice as big..."*).

Furthermore, by using different datasets, we have shown that different DR techniques do a better job in terms of explainability between clusters. However, another limitation is that data might be structured in different ways which produces different types of clusters. It directly follows from our results that this explanation method heavily relies on the generated clusters, and more importantly, its shapes and variance in the latent space, which is not desirable.

In section 3.5, we opted for an approach in which the latent spaces are dynamically scaled for the purpose of achieving a certain amount of variance in the data, as to obey the fixed hyperparameter $\epsilon$ which was introduced in section 3.4. A more robust method would be to dynamically define $\epsilon$ based on the variance in the latent space depending on the dataset and, potentially, proper handling of its outliers. We view the lack of expansion on this hyperparameter in the original paper as a (minor) limitation.

As VAEs are sampling points in latent space from a distribution, an identical point $x$ in the input space which is repeatedly mapped to latent space will yield different mappings. It would be interesting to further investigate to what extent explanatory algorithms can 'handle' this variance – to what extent such algorithms could find a stable and unchanging explanation in an ever-changing counterfactual world.

## 5.1  What was easy

After having asserted the dependencies needed to run the code provided by the authors on the GitHub repository, replicating the experiments performed in the paper was easy; the code was cleanly written, and it was easy to understand the architectural choices which the authors' made in their model.

## 5.2  What was difficult

Although the code required for computing the explanations was separated from the implementations in other folders in which it is applied on the different datasets, Plumb *et al*. decided to perform experiments using a VAE and intertwined all code for generating the explanations with the `scvis`-library. This caused the explanatory model to rely on `TensorFlow`, while the explanation method itself does not require any deep learning. It is preferred to dissect the model into independent components, considering that the explanatory model should work with other DR algorithms, including those which do not require neural pipelines (see section 3). Because of the authors' choice to use a VAE, the entire repository relied on `TensorFlow`. For that reason, we rewrote everything from scratch, to provide the dimensionality reduction methods on a stand-alone basis. As we were aiming to reproduce the results provided by the original paper, we also had to rewrite the external `scvis`-library, which provided the VAE architecture, in `PyTorch` (for the sake of keeping clean, separated and object-oriented code). This was very time-consuming as both frameworks are fundamentally different (dynamic vs static graph definition). The process of rewriting took approximately two weeks.

Lastly, results for the Bipolar could only be replicated with the provided model configuration file. Although we have retrained the model using the upgraded originally provided code, we have chosen to exclude this dataset for the experiments due to lack of computation power and time.

## 5.3  Communication with original authors

A short interaction occurred with the author of the paper, in order to gain more insight into the required external libraries as these were not listed in any documentation. Although the author responded very quickly, the issue had already been solved in the meantime.

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

# Appendices

## A   Additional latent spaces and corresponding measures

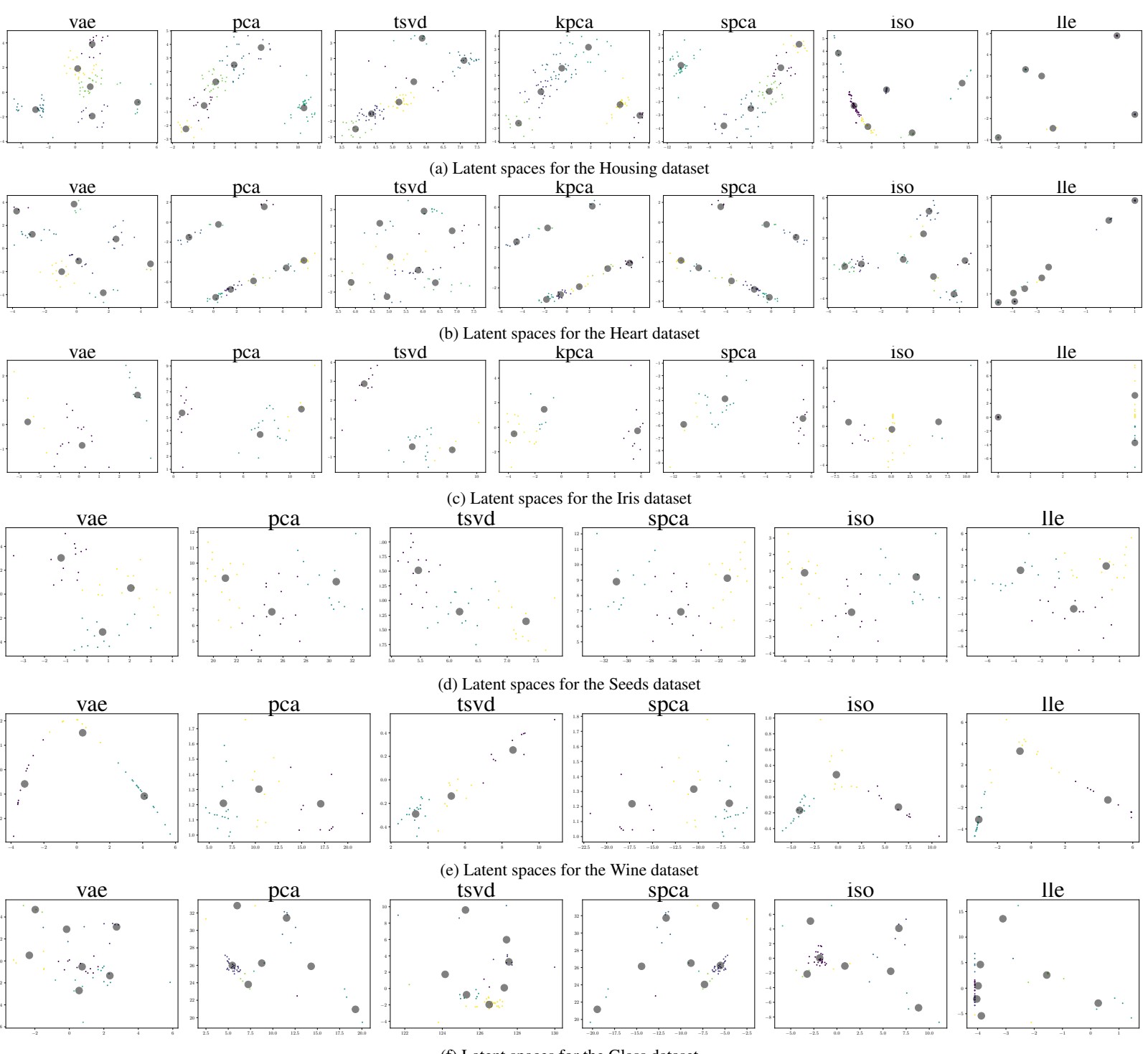

Figure 5: Visualization of two-dimensional latent spaces in which higher dimensional data is compressed after applying various dimensionality reduction techniques for different datasets. This figure illustrates how the underlying structure of the data, in combination with the involved method for reducing dimensionality, influences the resulting representational spaces. Every circle represents a cluster to which the (cluster-colored) data points belong.

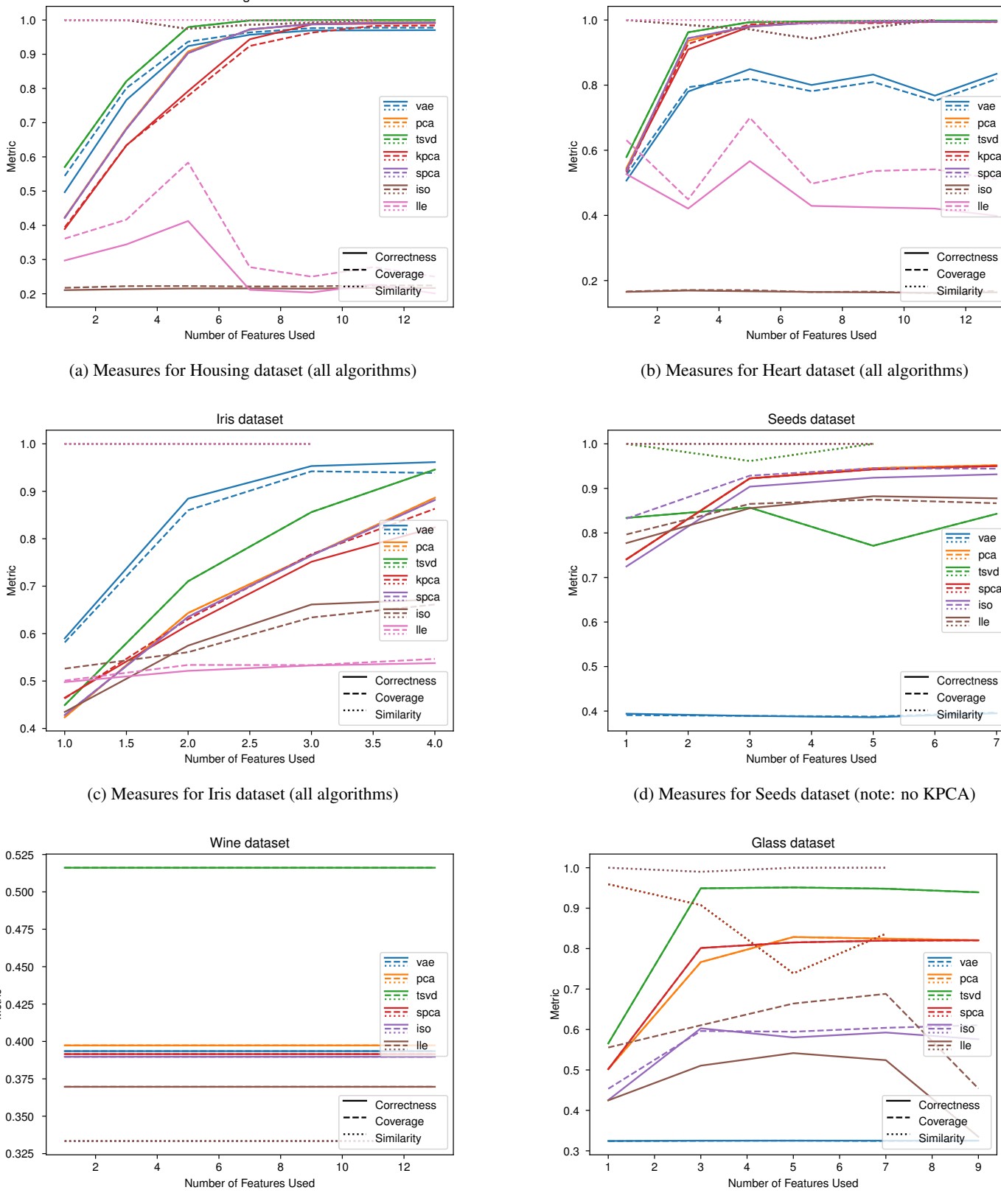

(a) Measures for Housing dataset (all algorithms)

(b) Measures for Heart dataset (all algorithms)

(c) Measures for Iris dataset (all algorithms)

(d) Measures for Seeds dataset (note: no KPCA)

(e) Measures for Wine dataset (note: no KPCA)

(f) Measures for Glass dataset (note: no KPCA)

Figure 6: Measures for coverage, correctness and similarity for all involved dimensionality reduction algorithms on all datasets.

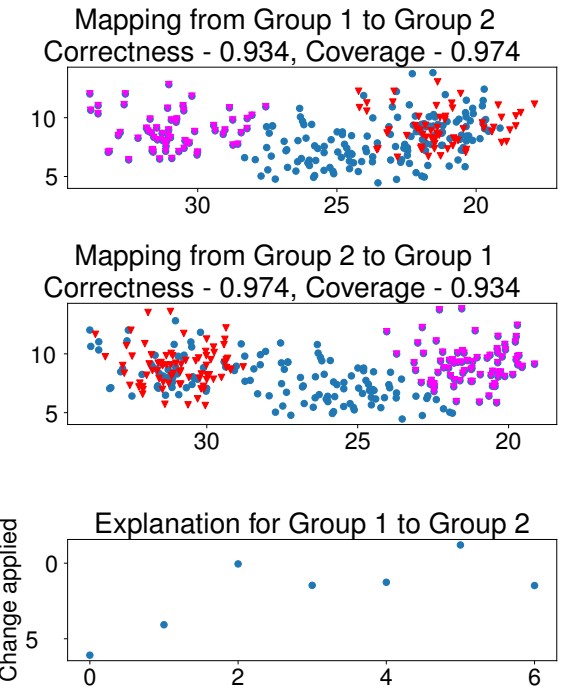

(a) A proper explanation for clusters in the Seeds dataset mapped to a latent space generated by SPCA.

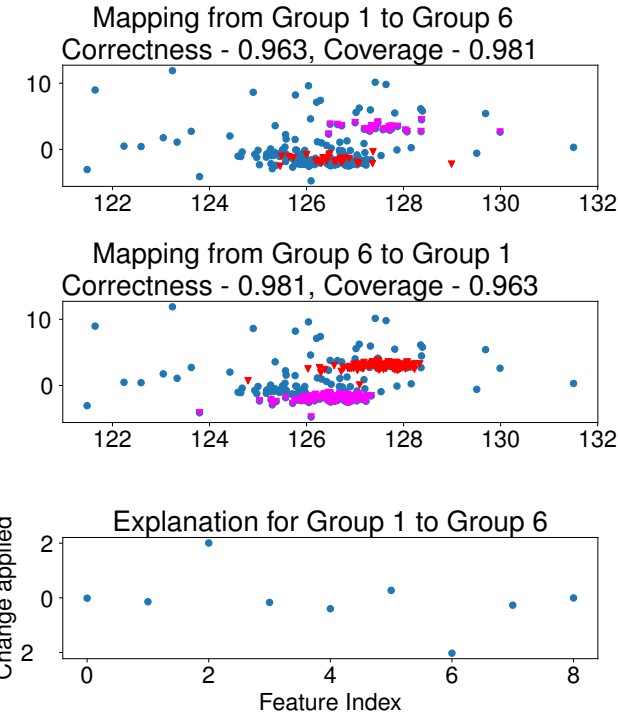

(b) A well-formed explanation for groups within the Glass dataset transformed by TSVD.

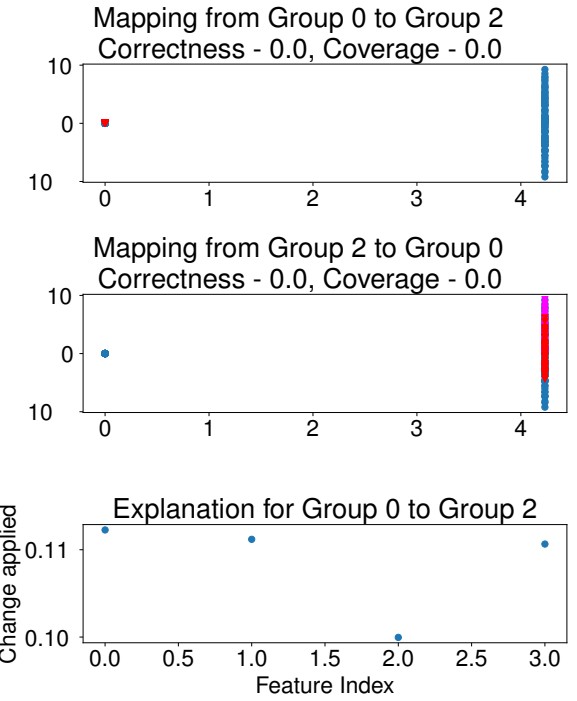

(c) Iris dataset mapped with LLE. Note how the model is unable to learn, caused by the shapes of the clusters.

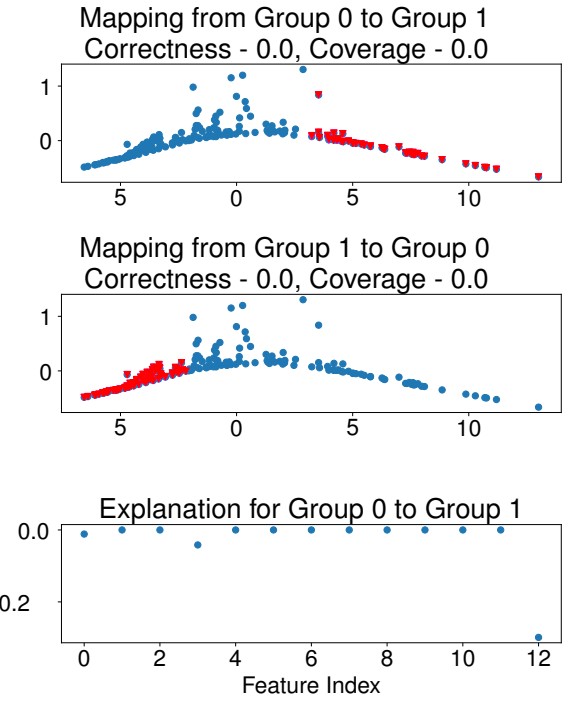

(d) Wine dataset mapped with an isomap. The model is unable to learn, indicating that a translation cannot suffice as an explanation here. Note that $k$-sparsity does not influence the explanation, as the Wine dataset has only one significant latent variable.

Figure 7: Four sample explanations between two arbitrary groups for four different datasets projected onto four different latent spaces.

