# OpenReview forum: "[Re] Explaining Groups of Points in Low-Dimensional Representations"
_ML_Reproducibility_Challenge/2020 — RC2020_

### Official Review · AnonReviewer1 · 2021-03-01
**A good report, results could be more clear and missing a few important information (how code was re-implemented, Hyperparameter selection).**

**Rating:** 7
**Confidence:** 4

**Review:**


# Reproducibility Summary
Main summary not clear. Is it the objective of the original paper, or the objective of the reproduction? The rest of the summary is clear.

# Scope of reproducibility
They reproduce results from original paper, and test with additional dimensionality reduction algorithms as well as on additional datasets.

# Code
Was extended with additional DR algorithms and re-implemented in PyTorch. It is not clearly mentioned however whether the algorithms were re-implemented solely based on information found in the paper or if the code has been used as a reference.

# Communication with original authors
No major communication, only to assert the dependencies needed to run the original cod.

# Hyperparameter Search
The hyperparameters are described but there is no mention of hyperparameter optimization.

# Ablation Study
The results are tested on additional DR algorithms. I would consider this as an ablation study since it verifies the effect of one of the components of the whole procedure.

# Discussion on results
The results are well described and compared with original work. The results section is more difficult to read than the rest of the report. The figures should be better described, it is very difficile to make sense of Figure 5 in particular


# Recommendations for reproducibility
None

# Results beyond the paper :
The method is tested with different DR and on 3 additional datasets. The comparison on the additional datasets is interesting because these have no low-dimensionality manifolds and thus should be trickier for the DR methods.

# Overall organization and clarity
The paper is very clear and especially well written with the exception of the results section. The figures lack explanation. See minor comments for grammatical errors.



# Comments

Section 3: Methodology
It should be made clear whether the re-implementation has been attempted with only looking at information available in the paper or if they used the available code-base as a reference. This makes an important difference for the reproduction as looking at the code-base could lead to re-implementation of important tricks that were not mentioned in the paper.

It would be good to cite the work behind the linear and non-linear models listed in 3.1.1 and 3.1.2.

I don’t understand the role of the digression. I feel it states in different words what has already been said implicitly in the prior sections.

Footnotes 14 and 15: I was clueless about what I should do to make the comparisons. Ideally these results should be integrated in the paper and presented clearly.

# Minor comments
Figure 1’s caption: I don’t understand the sentence: 'the difference between point 1 and 3 is explained each data point can be seen as an individual cluster. '

Section 3.5: we have ran -> we have run

Section 3.5: All experiments are ran -> All experiments were run

Footnote 13: be only using -> by only using

Section 5.1: set up was the obtaining > set up was obtaining

Section 5.2: Since we are aim -> Since we were aiming

**Familiar With The Original Paper:**

I have not read the original paper

**Reproducibility Summary:**

Report has summary

---

### Official Review · AnonReviewer3 · 2021-03-10
**Good Report exposing limitations but needs structuring**

**Rating:** 8
**Confidence:** 4

**Review:**

### SUMMARY
The paper claims that:
- The common data exploration workflow (of learning low dimensional representations, identifying features which help examine differences across clusters to determine what they represent as they correlate to an unobserved concept of interest) is treated as an interpretable machine learning problem where:
    - Global Counterfactual Explanations (GCE's) ensure pair-wise explanations for all points within a cluster
    - Transitive Global Translations (TGT's) generalize the above compressed sensing solution to find the complete set of explanations are both symmetrical and transitive among all groups simultaneously and empirically demonstrate the same with the following datasets: synthetic, UCI (Iris, Boston Housing and Heart Diseases data) as well as single-cell RNA data with adequate correctness and coverage
- TGT's identify explanations that accurately explain models while being relatively sparse and reportedly match underlying patterns in the data.

The submitted report addresses the above claims as follows:
- _Re-execution_ of existing code along with re-written code variants (upgraded to TF 2.x, Pytorch and without external dependencies such as on scvis) on all the above mentioned datasets establishes correctness, coverage, and sparsity, thus verifying both claims.
- Additionally, experimentation with other linear and non-linear dimensionality reduction algorithms (truncated SVD, sparse PCA, Gaussian variational autoencoder, kernel PCA, manifold dimensionality reduction algorithms like isomap and local linear embedding) on the following additional datasets -(seeds, Wine and Glass. dataset excluding single-cell RNA) - explored along with dynamic scaling of data in latent spaces for the purpose of achieving a certain amount of variance in order to test applicability to differing data structures, uncovers the following limitations:
    - Constrained variable freedom interferes in manifold mapping where matrix based explanations/explanations beyond translations (such as with rotation/scaling) may be necessary.
    - Structure of the data produces different type of clusters and hence, structure, shape, method of cluster annotation and variance in the latent space affects algorithmic performance.
     - Highly non-linear dimensionality reduction algorithms perform worse in terms of explainability (probably due to sparsity).

### MERITS
The additional experimentation is rather impressive and the report reflects an intuitive understanding of concepts such as coverage, correctness, and counterfactual explanations.

### MINOR CORRECTIONS
- In Methodology, "Exprimentation was done on a Macbook"; **Correction:** "_Experimentation_ was done on a Macbook"
- In Section 1 - Introduction, for the argument regarding algorithmic decisions that involve grouping of data, the reviewer recommends improving the context of the statement by clarifying settings (for instance,  the original paper discusses naturally arising grouping under the context of encoders or decoders) or by supporting this claim through citations.
- In Section 2 - Scope of Reproducibility, "do do not use the model"; **Correction:** "_do_ not use the model"
- In Section 3.1.1 - Linear Methods, "SPCA reduce the dimentionality in a linear fashion"; **Correction:** "SPCA reduce the _dimensionality_ in a linear fashion"
- In Section 3.1.2 - Non-Linear Methods , "While Isomap could be seen as a extension of KPCA, LLE can be understood as a combination of PCAs ran on local neighborhoods."; **Correction:** "While Isomap could be seen as _an_ extension of KPCA, LLE can be understood as a combination of PCAs _run_ on local neighborhoods"
- In Section 3.3 - Datasets, "sigmoidial Kernel-PCA procedure"; **Correction:** "_sigmoidal_ Kernel-PCA procedure"
- In Section 3.5 - Experimental setup and computational requirements, "Exprimentation was done on a Macbook"; **Correction:** "_Experimentation_ was done on a Macbook"
- In Section 5 - Discussion, "latent space will yield different different mappings"; **Correction:** "latent space will yield _different_ mappings or inconsistently different mappings"
- In Section 5.1 - What was easy, "set up was the obtaining the right version"; **Correction:** "set up was _obtaining_ the right version"
- Optionally, formatting of references can be enhanced: Explaining groups of points in low-dimensional representations -> Explaining Groups of Points in Low-Dimensional Representations.
- In general, the reviewer is of the opinion that the report could be structured in a more organized fashion:
    - In Section 5.1 - What was easy, it's recommended to move the discussions pertaining to "the hardest part" to Section 5.2.
    - Reduce the redundancy within the paper. For example, footnote 3 and footnote 1 essentially the same.
    - In Figure 1, it might be helpful to reuse the author's notations of ````````````` $R_{initial}$, $R_{target}$ representations and  $X_{initial}$, $X_{target}$ preimages etc.
    - For consistency, move all links to footnote. (For example: link in Section 3.5)
    - Reduce the back and forth between the report and paper. (For example: restate equation 9 in Section 3.4)

### RECOMMENDATIONS
- Please anonymize your submission. Also, note that your summary **must fit** in the first page.
- The original authors of the paper consider a **similarity metric** across explanations which has not been examined in this reproducibility report. Kindly address the same.
- Optionally, you could consider discussing the best set of hyperparameters from the experiments that resulted in reported results.
- Optionally, you could also demonstrate causal structures of the data and the inability of the [Plumb et al model](https://openreview.net/forum?id=MFj70_2-eY1) to capture the same (specifically with differently structured data, higher dimensional data or varying cluster variance, shape etc) to further strengthen your argument regarding shortcomings - similar to Table 1 of original paper.
- As per the [Machine Learning Reproducibility Checklist](https://www.cs.mcgill.ca/~jpineau/ReproducibilityChecklist.pdf), you could include statistics of the datasets in a tabular form and a tabulation of important results in the README file on [your github repository](https://github.com/giguru/fact-ai).
- As far as the creative insight on compressions is concerned, this space has been fairly explored before, [even in the context of compressed sensing](https://ermongroup.github.io/blog/uae/). I hence recommend moving the discussion to applications or perhaps reviewing this discussion in context of a specific dataset or application like, in the case of [complementary biological representations](https://genomebiology.biomedcentral.com/articles/10.1186/s13059-020-02021-3).
- For explanations beyond translations (x′=Mx+δ), the reviewer *appreciates the effort taken in this new direction* but critically, the argument of "incorporating rotation as an explanation that comes at the cost of explainability" sounds delicate.


**Familiar With The Original Paper:**

I have read the original paper

**Reproducibility Summary:**

Report has summary

---

### Decision · Program_Chairs · 2021-03-31

**Decision:**

Accept

**Comment:**

Selected for ReScience-C Journal Publication.

In addition to the results recreating the original paper, this reproduction extends the original work by further analyzing correctness and coverage, two ideas central to the claims in the paper. This work is clear and well-presented, and for example includes the hyperparameters that were used, which will be useful to those wishing to build upon this work.